# GRAPH LEARNING VIA SPECTRAL DENSIFICATION

## ABSTRACT

Graph learning plays an important role in many data mining and machine learning tasks, such as manifold learning, data representation and analysis, dimensionality reduction, data clustering, and visualization, etc. For the first time, we present a highly-scalable spectral graph densification approach (GRASPEL) for graph learning from data. By limiting the precision matrix to be a graph-Laplacian-like matrix in graphical Lasso, our approach aims to learn ultra-sparse undirected graphs from potentially high-dimensional input data. A very unique property of the graphs learned by GRASPEL is that the spectral embedding (or approximate effective-resistance) distances on the graph will encode the similarities between the original input data points. By interleaving the latest high-performance nearly-linear time spectral methods, ultrasparse yet spectrally-robust graphs can be learned by identifying and including the most spectrally-critical edges into the graph. Compared with prior state-of-the-art graph learning approaches, GRASPEL is more scalable and allows substantially improving computing efficiency and solution quality of a variety of data mining and machine learning applications, such as manifold learning, spectral clustering (SC), and dimensionality reduction.

## 1 INTRODUCTION

Graph learning is playing increasingly important roles in many machine learning and data mining applications. For example, a key step of many existing machine learning methods requires converting potentially high-dimensional data sets into graph representations: it is a common practice to represent each (high-dimensional) data point as a node, and assign each edge a weight to encode the similarity between the two nodes (data points). The constructed graphs can be efficiently leveraged to represent the underlying structure of a data set or the relationship between data points (Jebara et al., 2009; Maier et al., 2009; Liu et al., 2018). However, how to learn meaningful graphs from large data set at scale still remains a challenging problem.

Several recent graph learning methods leverage emerging graph signal processing (GSP) techniques for estimating sparse graph Laplacians, which show very promising results (Dong et al., 2016; Egilmez et al., 2017; Dong et al., 2019; Kalofolias & Perraudin, 2019). For example, (Egilmez et al., 2017) addresses the graph learning problem by restricting the precision matrix to be a graph Laplacian and maximizing a posterior estimation of attractive Gaussian Markov Random Field (GMRF) [1], while an $l1$-regularization term is used to promote graph sparsity; (Rabbat, 2017) provides an error analysis for inferring sparse graphs from smooth signals; (Kalofolias & Perraudin, 2019) leverages approximate nearest-neighbor (ANN) graphs to reduce the number of variables for optimization; (Kumar et al., 2019) introduces a graph Laplacian learning method by imposing Laplacian spectral constraints. However, even the state-of-the-art Laplacian estimation methods for graph learning do not scale well for large data set due to their extremely high algorithm complexity. For example, solving the optimization problem for Laplacian estimation in (Dong et al., 2016; Kalofolias, 2016; Egilmez et al., 2017; Dong et al., 2019) requires $O(N^2)$ time complexity per iteration for $N$ data entities and nontrivial parameters tuning for controlling graph sparsity which limits their applications to only very small data sets (e. g. with up to a few thousands of data points); the method introduced in (Carey, 2017) leverages Isomap manifold embedding (Tenenbaum et al., 2000) for graph construction, which requires $O(N^3)$ time for manifold construction and thus does not scale to large data set; the latest graph learning approach (Kalofolias & Perraudin, 2019) takes advantages of ANN graphs, but it still runs very slowly for large data sets; the Laplacian estimation method with

---

[1]If the precision matrix of a GMRF is an M-matrix with all non-negative off-diagonal elements, we call it an attractive GMRF (Slawski & Hein, 2015; Dong et al., 2019).

spectral constraints requires a good graph structure to be provided in advance (Kumar et al., 2019), which otherwise can be very costly when going through exhaustive graph structure searches.

This work for the first time introduces a *spectral graph densification* approach (GRASPEL) for learning ultra-sparse graphs from data by leveraging the latest results in spectral graph theory (Feng, 2016; 2018; Zhao et al., 2018). GRASPEL has a close connection with prior GSP-based Laplacian estimation methods (Dong et al., 2016; Kalofolias, 2016; Egilmez et al., 2017; Kalofolias & Perraudin, 2019; Dong et al., 2019), and the graphical Lasso method (Friedman et al., 2008). By treating $M$-dimensional data points as $M$ graph signals, GRASPEL allows efficiently solving a convex problem by iterative identifying and including the most spectrally-critical edges into the latest graph leveraging recent nearly-linear time spectral methods (Feng, 2016; 2018; Zhao et al., 2018). Compared with prior spectral graph sparsification algorithms (Spielman & Srivastava, 2011; Feng, 2016) that aim to remove edges from a given graph while preserving key graph spectral properties, GRASPEL aims to add edges into the graph such that the learned graphs will have spectral embedding (or effective-resistance) distances encoding the distances between the original input data points. Comparing with state-of-the-art graph learning methods, GRASPEL is more scalable for estimation of attractive Gaussian Markov Random Fields (GMRFs) for even very large data set. We summarize the contribution of this work as follows:

• We propose a spectral graph densification approach (GRASPEL) that allows efficient estimation of attractive Gaussian Markov Random Fields (GMRFs) leveraging the latest spectral graph theory.

• We show that the graphical Lasso problem with a Laplacian-like precision matrix can be efficiently solved by including spectrally-critical edges to dramatically reduce spectral embedding distortions.

• The key to achieving high efficiency is a spectral embedding scheme for finding spectrally-critical edges, allowing each GRASPEL iteration to be completed in $O(N \log N)$ instead of $O(N^2)$ time.

• For the first time, we introduce a novel convergence criterion for graph learning tasks based on **graph spectral stability**: when the maximum embedding distortion becomes small enough, or equivalently when graph spectra become sufficiently stable, GRASPEL iterations can be terminated.

• Our experiment results show that the graphs learned from high-dimensional data using GRASPEL can lead to more efficient and accurate spectral clustering (SC) as well as dimensionality reduction.

## 2 BACKGROUND OF GRAPH LEARNING VIA LAPLACIAN ESTIMATION

Given $M$ observations on $N$ data entities in a data matrix $X = [x_1, ..., x_M] \in \mathbb{R}^{N \times M}$, each column vector of $X$ can be considered as a signal on a graph. For example, the USPS data set including $9,298$ images of handwritten digits with each image having $256$ pixels will result in a feature matrix $X \in \mathbb{R}^{N \times M}$ with $N = 9,298$ and $M = 256$. The recent GSP-based graph learning methods (Dong et al., 2016) estimate graph Laplacians from $X$ for achieving the following desired characteristics:

**Smoothness of Graph Signals.** The graph signals corresponding to the real-world data should be sufficiently smooth on the learned graph structure: the signal values will only change gradually across connected neighboring nodes. The smoothness of a signal $x$ over an undirected graph $G = (V, E, w)$ can be measured with the following Laplacian quadratic form:

$$x^\top L x = \sum_{(p,q) \in E} w_{p,q} (x(p) - x(q))^2, \tag{1}$$

where $L = D - W$ denotes the Laplacian matrix of graph $G$ with $D$ and $W$ denoting the degree and the weighted adjacency matrices of $G$, and $w_{p,q}$ denotes the weight for edge $(p, q)$. The smaller value of (1) indicates the smoother signals across the graph. To quantify the smoothness ($Q$) of a set of signals $X$ over graph $G$, the following matrix trace can be computed (Kalofolias, 2016):

$$Q(X, L) = \text{Tr}(X^\top L X), \tag{2}$$

where $\text{Tr}(\bullet)$ denotes the matrix trace.

**Sparsity of the Estimated Graph (Laplacian).** Graph sparsity is another critical consideration in graph learning. One of the most important motivations of learning a graph is to use it for downstream data mining or machine learning tasks. Therefore, desired graph learning algorithms should allow better capturing and understanding the global structure (manifold) of the data set, while producing

sufficiently sparse graphs that can be easily stored and efficiently manipulated in the downstream algorithms, such as graph clustering, partitioning, dimension reduction, data visualization, etc. To this end, the graphical Lasso algorithm (Friedman et al., 2008) has been proposed to learn the structure in an undirected Gaussian graphical model using $l1$ regularization to control the sparsity of the precision matrix. Given a sample covariance matrix $S$ and a regularization parameter $\beta$, graphical Lasso targets the following convex optimization task:

$$\max_{\Theta} : \log \det(\Theta) - \text{Tr}(\Theta S) - \beta \|\Theta\|_1, \tag{3}$$

over all non-negative definite precision matrices $\Theta$. The first two terms together can be interpreted as the log-likelihood under a Gaussian Markov Random Field (GMRF). $\| \bullet \|$ denotes the entry-wise $l1$ norm, so $\beta \|\Theta\|_1$ becomes the sparsity promoting regularization term. This model tries to learn the graph structure by maximizing the penalized log-likelihood. When the sample covariance matrix $S$ is obtained from $M$ i.i.d ( independent and identically distributed) samples $X = [x_1, ..., x_M]$ where $X \sim N(0, S)$ has an $N$-dimensional Gaussian distribution with zero mean, each element in the precision matrix $\Theta_{i,j}$ encodes the conditional dependence between variables $X_i$ and $X_j$. For example, $\Theta_{i,j} = 0$ implies that the corresponding variables $X_i$ and $X_j$ are conditionally independent, given the rest. However, the log-determinant problems are very computationally expensive. The emerging GSP-based methods infer the graph by adopting the criterion of signal smoothness (Kalofolias, 2016; Dong et al., 2016; Egilmez et al., 2017; Kalofolias & Perraudin, 2019). However, their extremely high complexities do not even allow for learning graphs with even a few thousands of nodes . Furthermore, these methods require nontrivial parameters tuning for achieving good performance.

## 3 GRASPEL: GRAPH SPECTRAL LEARNING AT SCALE

Similar to recent GSP-based Laplacian estimation methods, GRASPEL aims to more efficiently solve the following convex problem for estimation of attractive GMRFs (Dong et al., 2019; Lake & Tenenbaum, 2010) that is also similar to the graphical Lasso problem (Friedman et al., 2008):

$$\max_{\Theta} : \log \det(\Theta) - \frac{1}{M} \text{Tr}(X^\top \Theta X) - \beta \|\Theta\|_1, \tag{4}$$

where $\Theta = L + \frac{1}{\sigma^2} I$, $L$ denotes the set of valid graph Laplacian matrices, $I$ denotes the identity matrix, and $\sigma^2 > 0$ denotes prior feature variance. It can be shown that the three terms in (4) are corresponding to $\log \det(\Theta)$, $Tr(\Theta S)$ and $\beta \|\Theta\|_1$ in (3), respectively. When each column vector in the data matrix $X$ [2] is treated as a graph signal vector, there is a close connection between our formulation and the graphical Lasso problem. Since $\Theta = L + \frac{1}{\sigma^2} I$ correspond to symmetric and positive definite (PSD) matrices (or M matrices) with non-positive off-diagonal entries, this formulation will lead to the estimation of attractive GMRFs (Dong et al., 2019; Slawski & Hein, 2015). In case that $X$ is non-Gaussian, (4) can be understood as Laplacian estimation based on minimizing the Bregman divergence between positive definite matrices induced by the function $\Theta \mapsto -\log \det(\Theta)$ (Slawski & Hein, 2015).

### 3.1 THEORETICAL BACKGROUND

Express the Laplacian matrix as

$$L = \sum_{(p,q) \in E} w_{p,q} e_{p,q} e_{p,q}^\top \tag{5}$$

where $e_p \in \mathbb{R}^N$ denotes the standard basis vector with all zero entries except for the $p$-th entry being 1, and $e_{p,q} = e_p - e_q$. Consider the objective function $F$ in (4):

$$F = \log \det(\Theta) - \frac{1}{M} Tr(X^\top \Theta X) - \beta \|\Theta\|_1, \tag{6}$$

which can be further simplified as follows by substituting (5) into (6):

$$F = \sum_{i=1}^{N} \log(\lambda_i + 1/\sigma^2) - \frac{1}{M} \left( \frac{Tr(X^\top X)}{\sigma^2} + \sum_{(p,q) \in E} w_{p,q} \|X^\top e_{p,q}\|_2^2 \right) - 4\beta \sum_{(p,q) \in E} w_{p,q}, \tag{7}$$

---

[2] For each of the $N$ row vectors $X(i, :) \in \mathbb{R}^{1 \times M}$ where $i = 1, ..., N$, the following two-step data preprocessing will be performed: **(1)** $X(i, :) = X(i, :) - \mu_i$, where $\mu_i$ denotes the sample mean of $X(i, :)$; then $S = \frac{XX^\top}{M}$, which results in $\text{Tr}(\Theta S) = \frac{1}{M} \text{Tr}(X^\top \Theta X)$. **(2)** Feature normalization by $X = X/\|X\|_2$ .

where the Laplacian eigenvectors corresponding to the ascending eigenvalues $\lambda_i$ are denoted by $u_i$ for $i = 1, ..., N$, satisfying:

$$Lu_i = \lambda_i u_i. \tag{8}$$

Taking the partial derivative with respect to the weight $w_{p,q}$ of edge $(p, q)$ leads to:

$$\frac{\partial F}{\partial w_{p,q}} = \sum_{i=2}^{N} \frac{1}{\lambda_i + 1/\sigma^2} \frac{\partial \lambda_i}{\partial w_{p,q}} - \frac{1}{M}\|X^\top e_{p,q}\|_2^2 - 4\beta, \tag{9}$$

Since the last two terms in (9) are all fixed (constant) values for a given data matrix $X$ where $\beta$ can be considered as an additional offset added to all data pairs (candidate edges), we can drop the third term by simply setting $\beta = 0$, which will not impact the ranking of candidate edges in graph learning. The above simplification implies the second term alone will effectively penalize graph density for estimating Laplacian-like precision matrix: including more edges will result in a greater trace $Tr(X^\top \Theta X)$. The spectral perturbation analysis in Theorem 1 (Appendix) will lead to:

$$\frac{\partial \lambda_i}{\partial w_{p,q}} = \left(u_i^\top e_{p,q}\right)^2. \tag{10}$$

If we construct a subspace matrices $U_r$ for spectral graph embedding by using the first $r-1$ weighted nontrivial Laplacian eigenvectors as follows:

$$U_r = \left[\frac{u_2}{\sqrt{\lambda_2 + 1/\sigma^2}}, ..., \frac{u_r}{\sqrt{\lambda_r + 1/\sigma^2}}\right], \tag{11}$$

then by setting $\beta = 0$, (9) can be rewritten as follows:

$$\frac{\partial F}{\partial w_{p,q}} \approx \|U_r^\top e_{p,q}\|_2^2 - \frac{1}{M}\|X^\top e_{p,q}\|_2^2 = z_{p,q}^{emb} - \frac{1}{M}z_{p,q}^{data}, \tag{12}$$

where $z_{p,q}^{emb} = \|U_r^\top e_{p,q}\|_2^2$ and $z_{p,q}^{data} = \|X^\top e_{p,q}\|_2^2$ denote the $\ell_2$ distances in the spectral embedding space and the original data space, respectively. We note that using a larger $r$ for constructing $U_r$ will allow more accurate estimation of $\frac{\partial F}{\partial w_{p,q}}$, but at a higher computational cost.

## 3.2 GRAPH LEARNING VIA SPECTRAL DENSIFICATION

If a candidate edge $(p, q)$ has a relatively large $\frac{\partial F}{\partial w_{p,q}}$ value, it is considered as a **spectrally-critical edge**, since it will significantly impact (7) and have large spectral embedding distortions defined as

$$\eta_{p,q} = M\frac{z_{p,q}^{emb}}{z_{p,q}^{data}}, \tag{13}$$

Subsequently, (12) can be further simplified as follows:

$$\frac{\partial F}{\partial w_{p,q}} \approx \left(1 - \frac{1}{\eta_{p,q}}\right)z_{p,q}^{emb}, \tag{14}$$

which implies that including the candidate edges with greater $\eta_{p,q}$ and $z_{p,q}^{emb}$ values into the latest graph will allow a faster convergence of (4) in gradient descent (GD), as long as $\eta_{p,q} > 1$ holds.

**The proposed approach.** When $\sigma^2 \to +\infty$ and $r \to N$, the spectral embedding distance $z_{p,q}^{emb}$ becomes the effective-resistance distance $R_{p,q}^{eff}$, and the scaled spectral embedding distortion $\frac{\eta_{p,q}}{M} = w_{p,q}R_{p,q}^{eff}$ becomes the edge leverage score for spectral graph sparsification (Spielman & Srivastava, 2011) if each edge weight is computed by $w_{p,q} = \frac{1}{z_{p,q}^{data}}$. Prior research proves that for every undirected graph a spectrally sparsified graph with $O(N\log N)$ edges can be computed by sampling each edge according to its effective resistance (Spielman & Srivastava, 2011); on the contrary, GRASPEL can be regarded as a **spectral graph densification** procedure that aims to include $O(N\log N)$ spectrally-critical edges with large spectral embedding distortions. Therefore, the graphical Lasso problem with Laplacian-like precision matrix can be efficiently solved via a spectral graph densification procedure that iteratively includes the most spectrally-critical edges into the latest graph so that the distortion in the graph spectral embedding space can be drastically mitigated.

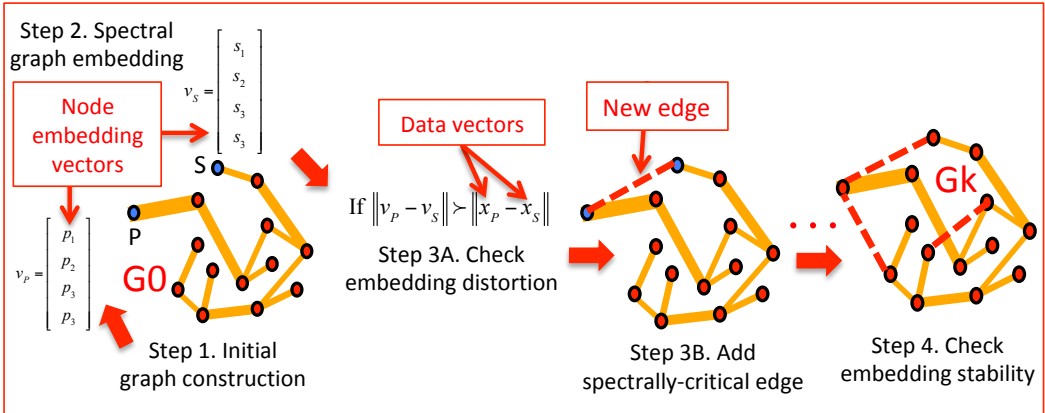

Figure 1: The overview of the proposed GRASPEL framework.

Consequently, a unique feature of GRASPEL is that the spectral embedding (effective-resistance) distances on the learned graph will encode the $\ell_2$ distances between the original data points, which is key to many manifold learning and dimensionality reduction problems (Belkin & Niyogi, 2003; Carey, 2017). We note that the proposed GRASPEL algorithm shares some similar features as the general stagewise algorithm (Tibshirani, 2015): as the step size goes to zero the sequence of forward stagewise estimates will exactly coincide with the lasso path. Consequently, GRASPEL will produce an approximate solution to the original graphical Lasso problem when using a rather small step size (e.g. adding only one edge with a small edge weight in each GRASPEL iteration).

**Convergence analysis.** The global optimal solution can be obtained when (14) becomes zero or there exists no edge with $\eta > 1$ for inclusion to the latest graph. Consequently, the convergence of GRASPEL iterations can be determined based on **graph spectral stability**: when the maximum embedding distortion $\eta$ becomes small enough (e.g. $\eta_{max} \leq tol$), or equivalently when graph spectra become sufficiently stable, the GRASPEL iterations can be terminated.

**Complexity analysis.** Comparing with the state-of-the-art graph construction methods (Dong et al., 2016; Kalofolias, 2016; Egilmez et al., 2017; Dong et al., 2019; Kalofolias & Perraudin, 2019) which require at least $O(N^2)$ time in each iteration, (14) allows each GRASPEL iteration to identify the most spectrally-critical edges in $O(N \log N)$ time: given the subspace projection matrix $U_r$ in (12) that can be computed in nearly-linear time leveraging our latest Laplacian eigensolver (Zhao et al., 2018), the spectral embedding distortion $(\eta_{p,q})$ of each candidate edge $(p,q)$ can be estimated in constant time; then (11) allows identifying the most spectrally-critical edges in almost nearly-linear time by limiting the search within only the candidate edge connections between a small number of the top and bottom sorted nodes according to the 1-D spectral embedding using the Fiedler vector.

## 4 DETAILED STEPS IN GRASPEL

**Overview.** GRASPEL will iteratively identify and add the most spectrally-critical edges into the latest graph so that the spectral embedding distortion can be greatly mitigated, until no such edges can be found (as illustrated in Figure 1). The detailed GRASPEL algorithm flow for graph learning has been described in Algorithm 1 in the Appendix, and summarized into the following key steps:

- **Step 1: Initial graph construction.** GRASPEL first constructs an ultra-sparse kNN graph to approximate the local manifold of a given data set. An extra step of spectral sparsification (Spielman & Srivastava, 2011; Feng, 2019) may be adopted to further simplify the graph.

- **Step 2: Spectral graph embedding.** GRASPEL extracts low-dimensional vector representations (e.g. $v_s$ for node $S$ in Figure 1) using scalable spectral graph embedding.

- **Step 3: Spectrally-critical edge identification.** GRASPEL identifies the most spectrally-critical edges with the greatest distortions, and include them into the latest graph.

- **Step 4: Spectral stability checking.** The GRASPEL iterations are terminated when the embedding distortions become sufficiently small, or equivalently when the graph spectra (e.g. the first few Laplacian eigenvalues and eigenvectors) become adequately stable.

**Step 1: Initial graph construction.** As aforementioned, (approximate) kNN graphs can be constructed as the initial graph, since they can be obtained efficiently (Muja & Lowe, 2009), while being able to approximate the local data proximity (Roweis & Saul, 2000). However, the optimal $k$ value (the number of nearest neighbors) is usually problem dependent and can be very difficult to find. In this work, GRASPEL will start with creating an (approximate) kNN graph using a relatively small $k$ value (e.g. $k = 2$), which will suffice for approximating the local structure of the manifold, and strive to iteratively improve the approximation of the global manifold structure by adding a small portion of spectrally-critical edges through solving the proposed convex problem in (4). In addition, spectral sparsification (Spielman & Srivastava, 2011; Feng, 2019) can be leveraged to further simplify the initial kNN graph (Wang & Feng, 2017).

**Step 2: Spectral graph embedding.** Spectral graph embedding directly leverages the first few nontrivial eigenvectors for projecting nodes onto low-dimensional space (Belkin & Niyogi, 2003). The eigenvalue decomposition of Laplacian matrix is usually the computational bottleneck in spectral graph embedding, especially for large graphs (Shi & Malik, 2000; Von Luxburg, 2007; Chen et al., 2011). To achieve good scalability, nearly-linear time Laplacian solvers (Koutis et al., 2010) or multilevel Laplacian solvers (Zhao et al., 2018) can be exploited for much faster eigenvector (eigenvalue) computations without loss of accuracy.

**Step 3: Spectrally-critical edge identification.** Once Laplacian eigenvectors are available for the latest graph, through the following phases GRASPEL will identify spectrally-critical edges by looking at each candidate edge's embedding distortion defined in (13), while the following theorem can be derived for quantifying each candidate edge's impact on the first few Laplacian eigenvalues.

**Theorem 1** *The perturbation of the $i$-th Laplacian eigenvalue $\lambda_i$ and the total relative spectral perturbation of the first $r$ Laplacian eigenvalues due to the inclusion a candidate edge $(p, q)$ can be estimated by $\delta\lambda_i = \delta w_{p,q} \left(u_i^\top e_{p,q}\right)^2$ and $\Delta_r = \sum\limits_{i=2}^{r} \frac{\delta\lambda_i}{\lambda_i} = \delta w_{p,q} \|U_r^\top e_{p,q}\|_2^2 \propto \eta_{p,q}$, respectively.*

Proof: See the Appendix.

*Phase A: candidate edge identification with Fiedler vectors.* Our approach for identifying spectrally-critical edges starts with sorting nodes according to the Fiedler vector that can be computed in nearly-linear time leveraging fast Laplacian solvers (Koutis et al., 2010; Spielman & Teng, 2014). This scheme is equivalent to including only the first nontrivial Laplacian eigenvector into $U_r$ $(r = 2)$ in (11) for spectral graph embedding. Subsequently, we can search candidate edge connections between the top and bottom few nodes in the 1D sorted node array. According to (14), only a small portion of node pairs with large embedding distances needs to be examined as candidate edges.

*Phase B: embedding distortion estimation with multiple eigenvectors.* With the first $r$ Laplacian eigenvectors computed in the previous spectral embedding step, each node of the latest graph will be associated with an $r$-dimensional embedding vector, which allows the spectral embedding distortion of each candidate (spectrally-critical) edge to be quickly estimated. As discussed in Section 3.2, the spectral embedding distances computed with the first $r$ eigenvectors can well approximate the effective-resistance distance thus the gradient in the proposed optimization task (9). Only the candidate edges with top embedding distortions will be added into the latest graph. Since the distances approximated using the first few eigenvectors will be the lower bounds of the effective-resistance distances, the lower bound of embedding distortions due to such an approximation can be estimated.

**Step 4: Spectral stability checking.** In this work, we propose to evaluate the edge embedding distortions with (13) for checking the spectral stability of the learned graph. If there exists no additional edge that has an embedding distortion greater than a given tolerance level ($tol$), GRASPEL iterations can be terminated. It should be noted that choosing different tolerance levels will result in graphs with different densities. For example, choosing a smaller distortion tolerance will require more edges to be included so that the resultant spectral embedding distances on the learned graph can more precisely encode the distances between the original data points. In practice, even for very large data set GRASPEL converges very quickly when starting with an initial 2NN or uNN (ultra-sparse nearest-neighbor) graph (Wang & Feng, 2017).

## 5 EXPERIMENTS

In this section, extensive experiments have been conducted to evaluate the performance of GRASPEL for a variety of public domain data sets (see the Appendix A.3 for detailed setting and evaluation metrics). In Sections A.6.1 and A.6.2 (see Appendix), we report additional experimental results for spectral clustering and dimensionality reduction (tSNE) applications leveraging the proposed GRASPEL approach. Since this work primarily focuses on learning graphs from high-dimensional data points, the proposed method can be orthogonal to existing research related to deep learning based spectral clustering methods: an autoencoder can be first applied to transform the input data into more optimal features that can subsequently become the input of GRASPEL for learning graphs in spectral clustering. For fair comparisons with other state-of-the-art graph learning methods, **we directly use the raw data as input without any additional pre-processing steps**. The following experiments are performed using MATLAB R2020b running on a Laptop with 10th Intel(R) Core(TM) i5 CPU and 8GB RAM.

### 5.1 EXPERIMENT SETUP

When applying Algorithm 1 to our data sets for the graph learning tasks shown in this section, we randomly sample candidate edges that connect between the top and bottom $0.05|V|$ ($\epsilon = 0.05$) nodes in the 1D sorted array according to the Fiedler vector, which allows GRASPEL to quickly identify the most spectrally-critical edges. Note that choosing a smaller $\epsilon$ value will allow more efficient edge sampling for estimating global graph (manifold) structural properties, while choosing a greater $\epsilon$ value will require more samples but may lead to better preservation of mid-to-short range graph (manifold) structural properties. When estimating the spectral distortion of each candidate edge we compute the first $r = 2$ to 10 Laplacian eigenvectors for the spectral graph embedding step. We set the edge sampling ratio to be $\zeta = 0.001$ and add one edge ($s = 1$) to the latest graph in each GRASPEL iteration. $\sigma = 1E3$ in (11) has been used for computing $\Theta$ in all experiments. Clustering Accuracy (ACC) and Normalized Mutual Information (NMI) values (see definitions in the Appendix) are used for evaluating the spectral clustering results.

### 5.2 EXPERIMENT RESULTS

**Spectral stability checking.** In Figure 2, we show how the objective function defined in (6) would change during the GRASPEL iterations when starting with **(a)** a 2NN graph, and **(b)** an ultra-sparse nearest neighbor (uNN) graph for the USPS data set. Only the first 50 Laplacian eigenvalues are computed for evaluating (6). The uNN graph is obtained by spectrally sparsifying a 5NN graph using the GRASS algorithm (with a relative condition number of 30) (Feng, 2019). We set the distortion tolerance to be $tol = 1$ for both cases and demonstrate the results for the first 30 GRASPEL iterations. As observed in Figure 2, (6) grows rapidly within the first 10 iterations when starting with the 2NN graph, which indicates the GRASPEL iterations allows quickly converging to a graph with relatively stable graph spectra [3]; **(b)** achieves a greater objective function value after 30 iterations when comparing with **(a)**, which is mainly due to the much sparser structure of the initial uNN graph. Therefore, for very large data sets a spectral sparsification procedure can be performed before going through GRASPEL iterations for further improving graph sparsity and thus runtime effi-

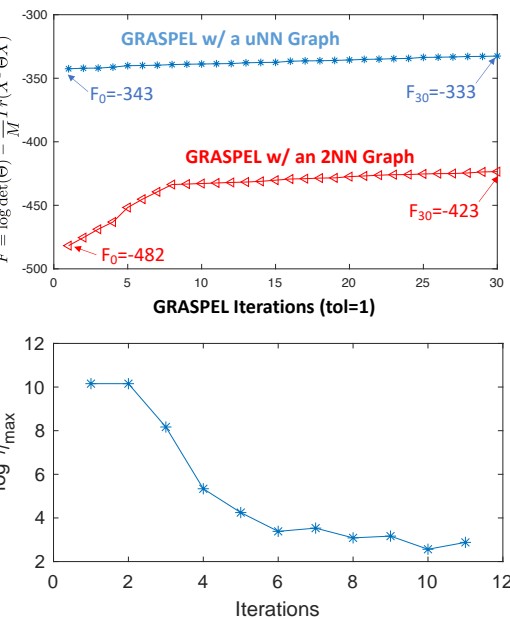

Figure 2: Objective function changes (top figure) and maximum spectral distortions (bottom figure) during GRASPEL iterations.

---

[3]GRASPEL starts to experience negligible gradient values after 10 iterations and can not identify additional spectrally-critical candidate edges with large embedding distortions.

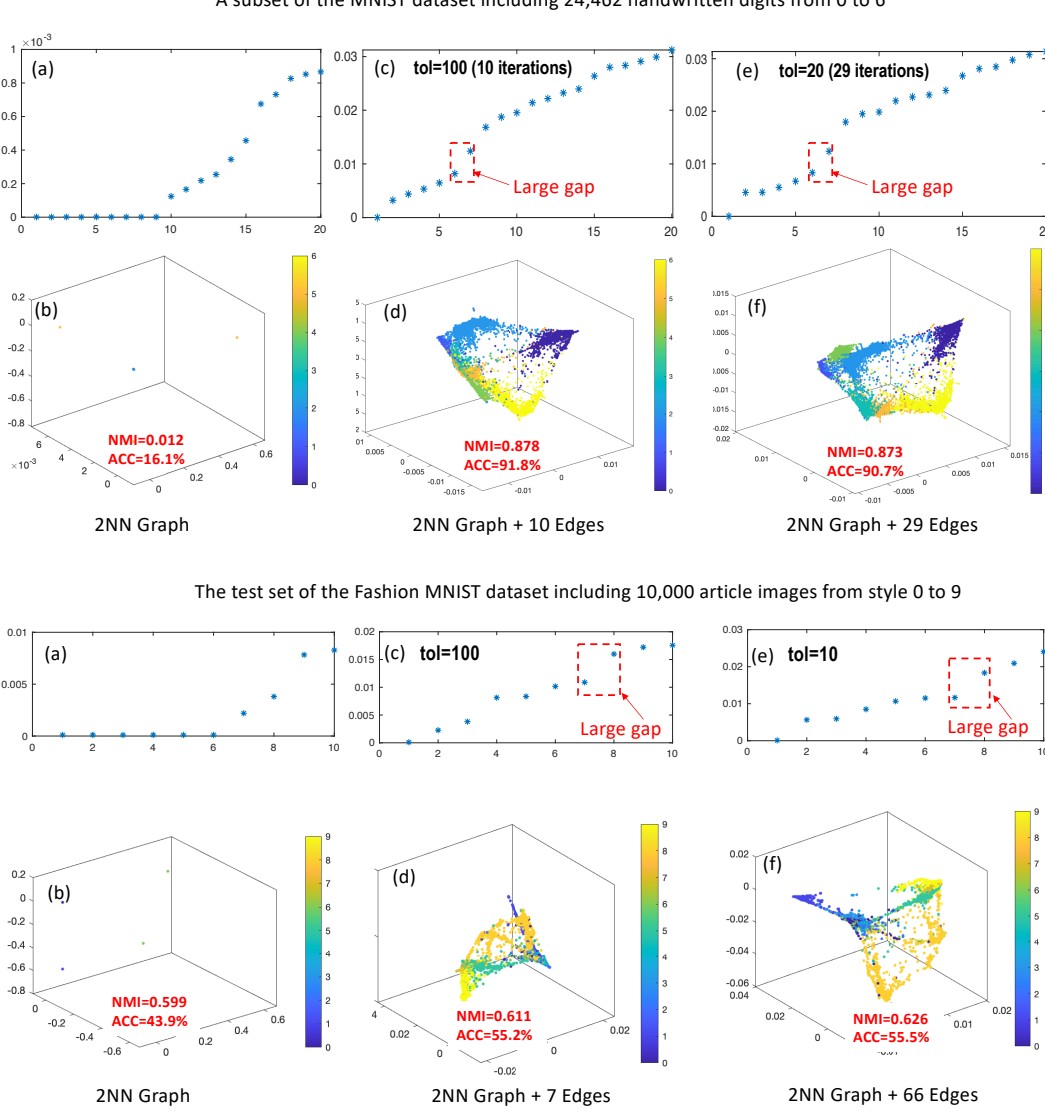

Figure 3: The first few Laplacian eigenvalues and 3D spectral drawings of the 2NN graph in figures (a) and (b), and the GRASPEL-learned graphs in figures (c) to (f).

ciency in graph learning tasks. However, it may be challenging to choose a good $k$ for creating the initial kNN graph that will be further spectrally sparsified: choosing a too large $k$ will result in very dense initial graph even after spectral sparsification, whereas choosing a small $k$ may lead to slower convergence.

**Spectral embedding distortion.** We also show how the proposed scheme for spectral stability checking can be applied based on the embedding distortion metric defined in (13). GRASPEL iterations can be terminated when there exists no candidate edge that has a spectral embedding distortion greater than a given tolerance level (e.g. $\eta \geq tol$). As shown in Figure 2, for the USPS data set GRASPEL adds one additional edge in each iteration and requires only 11 iterations to effectively mitigate the maximum embedding distortion by over $2,600\times$.

**Spectral embedding results.** In Figure 3, we show the first few Laplacian eigenvalues and 3D spectral drawings of the graphs learned with different distortion tolerance levels for a subset (including all the $24,462$ handwritten digits from $0$ to $6$) of the MNIST data set (see Appendix for details) and the test set of the Fashion MNIST data set including $10,000$ article images from style $0$ to $9$ (Xiao et al., 2017). When creating the 3D spectral drawing layouts, each entry of the first

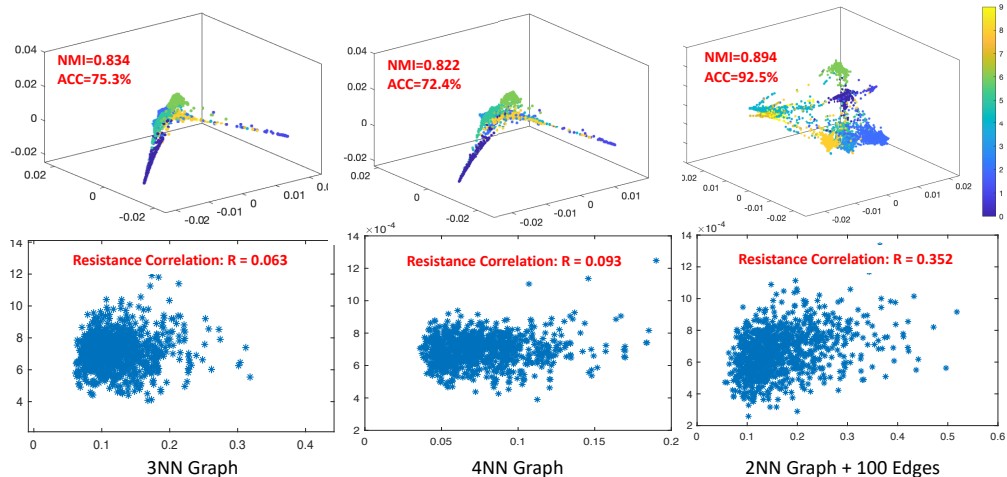

Figure 4: The Pearson correlation coefficients computed with the effective-resistance distances obtained on the 3NN graph (left), 4NN graph (middle) and the graph learned by GRASPEL (right).

three nontrivial Laplacian eigenvectors $(u_2, u_3, u_4)$ corresponds to the $x$, $y$ and $z$ coordinates of each node (data point), respectively. The ground-truth label of each data point is shown using its corresponding color. The edges have been omitted in the graph layouts to better reveal the structure of the data points (manifolds). Different embedding distortion tolerance levels are considered in our experiment. Starting from an initial 2NN graph, by adding one edge in each iteration GRASPEL has dramatically mitigated the spectral embedding distortions from by adding only a few additional edges into the initial 2NN graph. By examining the first few Laplacian eigenvalues, we notice that the initial 2NN graph of the MNIST data set has nine connected components (that equals the number of zero eigenvalues), while with 10 extra edges added via GRASPEL iterations, a well-connected graph can be formed for approximately preserving the structure of the original data set. For the MNIST data set we also observe relatively large gaps between the 6th and 7th eigenvalues, indicating that the intrinsic dimensionality of the GRASPEL-learned graphs (manifolds) is approximately five, whereas for the Fashion MNIST data set the intrinsic dimensionality is approximately six.

**Resistance-distance correlation.** In the last, for the full USPS data set we evaluate graph learning quality by checking if the effective-resistance distances on the learned graph will properly encode the $\ell_2$ distances between the original data points. To this end, we randomly pick up $1,000$ node pairs and compute their effective-resistance distances. To avoid picking up nearby nodes, we first sort nodes according to the Fiedler vector and then choose the node pairs $(n_{top}, n_{bot})$ from the node sets $N_{top}$ and $N_{bot}$ formed with the top and bottom $5\%|V|$ nodes, respectively. Then, these resistance distances are compared with the corresponding $\ell_2$ distances between the original data points by checking the Pearson correlation coefficient. In Figure 4, we observe that the effective-resistance distances on the graph learned via 100 GRASPEL iterations (27 seconds) have the highest correlation ($R = 0.352$) with the $\ell_2$ distance between the original data points, while the 3NN or 4NN graphs with much greater edge density only achieve $R = 0.063$ and $R = 0.093$, respectively. It is also observed that increasing $k$ from 3 to 4 for constructing the kNN graph can improve the resistance correlation but not necessarily the spectral clustering quality (e.g. NMI and ACC metrics), whereas the graph learned by GRASPEL achieves the best results in all aspects.

## 6 CONCLUSION

In this work, we present a highly-scalable *spectral graph densification* approach (GRASPEL) for graph learning from data. By limiting the precision matrix to be a graph-Laplacian-like matrix in graphical Lasso, GRASPEL can always efficiently learns sparse undirected graphs from high-dimensional input data by identifying and including the most spectrally-critical edges into the latest graph. A unique feature of the graphs learned by GRASPEL is that the effective-resistance distances will encode the similarities of the original data points. Compared with state-of-the-art graph learning approaches, GRASPEL is more scalable and always leads to substantially improved computing efficiency and solution quality for spectral clustering (SC) and dimensionality reduction tasks.

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

# A   APPENDIX

## A.1   PROOF OF THEOREM 1

Let $L \in \mathbb{R}^{N \times N}$ denote the Laplacian matrix of an undirected graph, and $u_i$ denote the $i$-th eigenvector of $L$ corresponding to the $i$-th eigenvalue $\lambda_i$ that satisfies:

$$Lu_i = \lambda_i u_i, \tag{15}$$

then we have the following eigenvalue perturbation analysis:

$$(L + \delta L)(u_i + \delta u_i) = (\lambda_i + \delta \lambda_i)(u_i + \delta u_i), \tag{16}$$

where a perturbation $\delta L = \delta w_{p,q} e_{p,q} e_{p,q}^\top$ that implies a new edge connection is applied to $L$, resulting in perturbed eigenvalues and eigenvectors $\lambda_i + \delta \lambda_i$ and $u_i + \delta u_i$ for $i = 1, ..., N$, respectively.

Keeping only the first-order terms leads to:

$$L\delta u_i + \delta L u_i = \lambda_i \delta u_i + \delta \lambda_i u_i. \tag{17}$$

Write $\delta u_i$ in terms of the original eigenvectors $u_i$ for for $i = 1, ..., N$:

$$\delta u_i = \sum_{i=1}^{N} \alpha_i u_i. \tag{18}$$

Substituting (18) into (17) leads to:

$$L \sum_{i=1}^{N} \alpha_i u_i + \delta L u_i = \lambda_i \sum_{i=1}^{N} \alpha_i u_i + \delta \lambda_i u_i. \tag{19}$$

Multiplying $u_i^\top$ to both sides of (19) results in:

$$u_i^\top L \sum_{i=1}^{N} \alpha_i u_i + u_i^\top \delta L u_i = \lambda_i u_i^\top \sum_{i=1}^{N} \alpha_i u_i + \delta \lambda_i u_i^\top u_i. \tag{20}$$

Since $u_i$ for for $i = 1, ..., N$ are unit-length, mutually-orthogonal eigenvectors, we have:

$$u_i^\top L \sum_{i=1}^{N} \alpha_i u_i = \alpha_i u_i^\top L u_i, \qquad \lambda_i u_i^\top \sum_{i=1}^{N} \alpha_i u_i = \alpha_i u_i^\top \lambda_i u_i. \tag{21}$$

Substituting (15) into (21), we have:

$$\alpha_i u_i^\top L u_i = \alpha_i u_i^\top \lambda_i u_i. \tag{22}$$

According to (21), we have:

$$u_i^\top \sum_{i=1}^{N} \alpha_i u_i = \lambda_i u_i^\top \sum_{i=1}^{N} \alpha_i u_i. \tag{23}$$

Substituting (23) into (20) leads to:

$$u_i^\top \delta L u_i = \delta \lambda_i u_i^\top u_i = \delta \lambda_i. \tag{24}$$

Then the eigenvalue perturbation $\delta \lambda_i$ due to $\delta L$ is given by:

$$\delta \lambda_i = \delta w_{p,q} \left( u_i^\top e_{p,q} \right)^2. \tag{25}$$

The total relative perturbation of the first $r$ eigenvalues due to the inclusion of edge $(p, q)$ becomes:

$$\Delta_r = \sum_{i=2}^{r} \frac{\delta \lambda_i}{\lambda_i} = \delta w_{p,q} \|U_r^\top e_{p,q}\|_2^2 = \delta w_{p,q} z_{p,q}^{emb}, \tag{26}$$

where $z_{p,q}^{emb}$ denotes the spectral embedding distance between $p$ and $q$ when $U_r$ is formed with the first $r - 1$ weighted nontrivial eigenvectors as defined in (11). If each edge weight is computed by $\delta w_{p,q} = \frac{1}{z_{p,q}^{data}}$ with $z_{p,q}^{data} = \|X^\top e_{p,q}\|_2^2$, as long as we can find a spectrally-critical edge with large $\Delta_r = \frac{z_{p,q}^{emb}}{z_{p,q}^{data}} = \frac{1}{M} \eta_{p,q}$, or equivalently spectral embedding distortion, including this edge into the current graph will significantly perturb the first $r - 1$ Laplacian eigenvalues.

## A.2 ALGORITHM FLOW

---

**Algorithm 1** The GRASPEL Algorithm Flow

---

**Input:** A data matrix ($X = [x_1, ... x_M] \in \mathbb{R}^{N \times M}$) with $N$ data points in $M$-dimensional, embedding distortion tolerance ($1 \leq tol$), window size for edge sampling ($0 < \epsilon \leq 50\%$), edge sampling ratio ($0 < \zeta \leq 1$), and the number of edges to be selected in each iteration ($0 < s$).
**Output:** The spectrally-learned graph $G$.

1: Construct an initial 2NN graph $G$ using approximate kNN algorithms.
2: **while** $\eta_{max} \geq tol$ **do**
3:  Embed the latest graph $G$ using its Fiedler vector and sort the nodes into a 1D array $I_{node}$;
4:  Obtain node set $N_{top}$ ( $N_{bot}$) by including only the top (bottom) $\lceil \epsilon N \rceil$ nodes in $I_{node}$;
5:  Sample each of the $\lceil s/\zeta \rceil$ edges by randomly choosing one node from $N_{top}$ and another node from $N_{bot}$;
6:  Form an edge set $E_{sel}$ using edges with large distortions ($\eta \geq tol$) and set the largest edge embedding distortion as $\eta_{max}$.
7:  **if** $|E_{sel}| \geq s$ **then**
8:    Add the top $s$ edges with largest $\eta$ from $E_{sel}$ into $G$;
9:  **else**
       Add all the edges in $E_{sel}$ into $G$;
10:  **end if**
11: **end while**
12: Return the learned graph $G$.

---

**Algorithm 2** Spectral Clustering Algorithm

---

**Input:** A graph $G = (V, E, w)$ and the number of clusters $r$.
**Output:** Clusters $C_1 ... C_r$.

1: Compute the adjacency matrix $A$, and diagonal matrix $D$;
2: Obtain the unnormalized Laplacian matrix $L = D-A$;
3: Compute the eigenvectors $u_1, ... u_r$ that correspond to the bottom $r$ nonzero eigenvalues of $L$;
4: Construct $U_r \in \mathbb{R}^{N \times r}$, with $r$ eigenvectors of $L$ stored as columns;
5: Perform k-means algorithm to partition the rows of $U_r$ into $r$ clusters and return the result.

---

## A.3 DATA SETS DESCRIPTION

COIL20: the data set contains $1,440$ gray-scale images of 20 objects, and each object on a turntable has 72 normalized gray-scale images taken from different degrees. The image size is 32x 32 pixels.

PenDigits: the data set consists of 7,494 images of handwritten digits from 44 writers, using the sampled coordination information. Each digit is represented by 16 attributes.

USPS: the data set includes $9,298$ scanned hand-written digits from 0 to 9 on the envelops from U.S. Postal Service with 256 attributes.

MNIST: the data set consists of 70,000 images of handwritten digits. Each image has 28-by-28 pixels in size. This database can be found at website (http://yann.lecun.com/exdb/mnist/).

A.4 COMPARED ALGORITHMS

**Standard kNN**: the most widely used affinity graph construction method. Each node is connected to its $k$ nearest neighbors.

**Consensus of kNN (cons-kNN)** (Premachandran & Kakarala, 2013): adopts the state-of-the-art neighborhood selection methods to construct the affinity graphs. It selects strong neighborhoods to improve the robustness of the graph by using the consensus information from different neighborhoods in a given kNN graph.

**LSGL** (Kalofolias & Perraudin, 2019): a method to automatically select the parameters of the model introduced in (Kalofolias, 2016) given a desired graph sparsity level. The default settings have been used in our experiments.

A.5 EVALUATION METRIC

(1) **The ACC metric** measures the agreement between the clustering results generated by clustering algorithms and the ground-truth labels. A higher value of $ACC$ indicates better clustering quality. The ACC can be computed by:

$$ACC = \frac{\sum\limits_{j=1}^{n} \delta(y_i, map(c_i))}{n},\tag{27}$$

where $n$ is the number of samples in the data set, $y_i$ is the ground-truth label provided by the data sets, and $c_i$ is clustering result obtained from the algorithm. $\delta(x,y)$ is a delta function defined as: $\delta(x,y)$=1 for $x = y$, and $\delta(x,y)$=0, otherwise. $map(\bullet)$ is a permutation function that maps each cluster index $c_i$ to a ground truth label, which can be realized using the Hungarian algorithm (Papadimitrou & Steiglitz, 1982).

**(2) The NMI metric** is in the range of [0, 1], while a higher NMI value indicates a better matching between the algorithm generated result and ground truth result. For two random variables $P$ and $Q$, normalized mutual information is defined as (Strehl & Ghosh, 2002):

$$NMI = \frac{I(P,Q)}{\sqrt{H(P)H(Q)}},\tag{28}$$

where $I(P,Q)$ denotes the mutual information between $P$ and $Q$, while $H(P)$ and $H(Q)$ are entropies of $P$ and $Q$. In practice, the NMI metric can be calculated as follows (Strehl & Ghosh, 2002):

$$NMI = \frac{\sum\limits_{i=1}^{k}\sum\limits_{j=1}^{k} n_{i,j} \log(\frac{n \cdot n_{i,j}}{n_i \cdot n_j})}{\sqrt{(\sum\limits_{i=1}^{k} n_i \log \frac{n_i}{n})(\sum\limits_{j=1}^{k} n_j \log \frac{n_j}{n})}},\tag{29}$$

where $n$ is the number of data points in the data set, k is the number of clusters, $n_i$ is the number of data points in cluster $C_i$ according to the clustering result generated by algorithm, $n_j$ is the number of data points in class $C_j$ according to the ground truth labels provided by the data set, and $n_{i,j}$ is the number of data points in cluster $C_i$ according to the clustering result as well as in class $C_j$ according to the ground truth labels.

A.6 ADDITIONAL EXPERIMENTAL RESULTS

A.6.1 GRAPH LEARNING FOR SPECTRAL CLUSTERING (SC)

The classical spectral clustering (SC) algorithm (see Algorithm 2 in the Appendix) first constructs a kNN graph where each edge weight encodes similarities between different data points (entities); then SC calculates the eigenvectors of the graph Laplacian matrix and embeds data points into low-dimensional space (Belkin & Niyogi, 2003); in the last, k-means algorithms are used to partition the data points into multiple clusters. The performance of SC strongly depends on the quality of the underlying graph (Guo, 2015). In this section, we apply GRASPEL for graph construction, and show the learned graphs can result in drastically improved efficiency and accuracy in SC tasks.

Since the SC algorithm has intrinsic randomness, the clustering result in each run is different. So the reported numbers in our results have been averaged over 20 runs.

We show comprehensive results of SC using four graph learning (construction) methods in Table 1 and Table 2. The runtime reported in Table 2 includes the total time for eigendecomposition of the Laplacian matrix and k-means clustering. The graph density ($|E|/|V|$) results are also shown in Table 3. In Table 4, the runtime of the consensus method includes the time for consensus information calculation and edge pruning, while the runtime of GRASPEL includes the total time for spectral graph densification. As observed, GRASPEL can consistently achieve the state-of-the-art results in SC: the GRASPEL learned graphs have led to significantly better SC accuracy (ACC), much lower graph densities and much shorter graph learning time.

Table 1: ACC and NMI results

| Data Set | ACC(%)/ NMI | | | |
| | Standard k-NN | Consensus | LSGL | GRASPEL |
|---|---|---|---|---|
| COIL20 | 75.72/0.86 | 81.60/0.90 | 85.49/**0.95** | **86.46**/0.94 |
| PenDigits | 74.36/0.79 | 71.08/**0.79** | 74.53/0.77 | **82.40/0.79** |
| USPS | 64.31/0.79 | 68.54/0.81 | 81.50/0.84 | **91.50/0.89** |
| MNIST | 64.20/0.74 | - | - | **74.63/0.78** |

Table 2: SC Runtime Results

| Data Set | Spectral clustering time (seconds) | | | |
| | Standard k-NN | Consensus | LSGL | GRASPEL |
|---|---|---|---|---|
| COIL20 | 0.03 | 0.03 | 0.08 | **0.02** |
| PenDigits | 0.18 | **0.16** | 4.42 | 0.17 |
| USPS | 0.72 | 0.56 | 7.05 | **0.28** |
| MNIST | 252.59 | - | - | **3.06** |

Table 3: Graph density results

| Data Set | Graph density ($|E|/|V|$) | | | |
| | Standard k-NN | Consensus | LSGL | GRASPEL |
|---|---|---|---|---|
| COIL20 | 6.12 | 5.06 | 11.99 | **1.39** |
| PenDigits | 6.76 | 6.70 | 186.52 | **2.96** |
| USPS | 7.30 | 6.58 | 29.97 | **1.70** |
| MNIST | 7.46 | - | - | **1.72** |

Table 4: Graph learning (construction) time results

| Data Set | Graph construction time (seconds) | | |
| | Consensus | LSGL | GRASPEL |
|---|---|---|---|
| COIL20 | 2.43 | 13.56 | **0.29** |
| PenDigits | 172.51 | 1085.43 | **2.04** |
| USPS | 574.28 | 2074.78 | **3.37** |
| MNIST | - | - | **208.89** |

In Figures 5 and 6, additional SC results have been provided for the USPS and the Pendigit data sets by comparing two embedding distortion tolerance levels ($tol = 100$ and $tol = 10$). Not surprisingly, when starting with initial 2NN graphs a few GRASPEL iterations have already dramatically improved SC results: the normalized mutual information (NMI) has been improved from $0.612$ to $0.888$ for the USPS data set, and from $0.04$ to $0.840$ for the Pendigit data set; The spectral clustering accuracy (ACC) has been improved from $40.4\%$ to $91.5\%$ for the USPS data set, and from $15.2\%$ to $89.4\%$ for the Pendigit data set.

Since the number of zero Laplacian eigenvalues equals to the connected components in the learned graph, we observe that GRASPEL can always identify $O(q)$ spectrally-critical edges added to the initial 2NN graph so that its $q$ connected components immediately get stitched into a connected graph for both test cases.

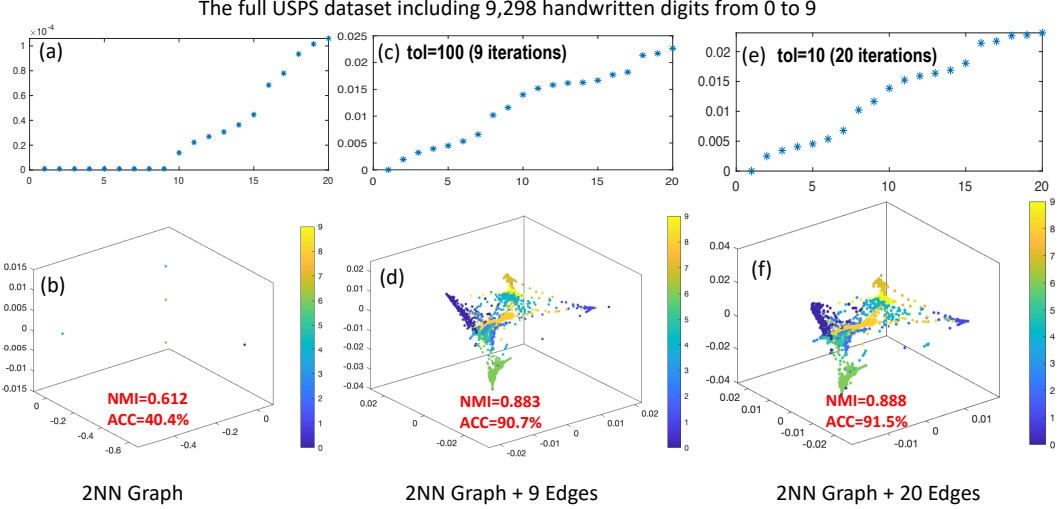

Figure 5: The first 20 Laplacian eigenvalues (top) and 3D spectral drawings (bottom) of the 2NN graph in figures (a) and (b), and the GRASPEL-learned graphs in figures (c) to (f).

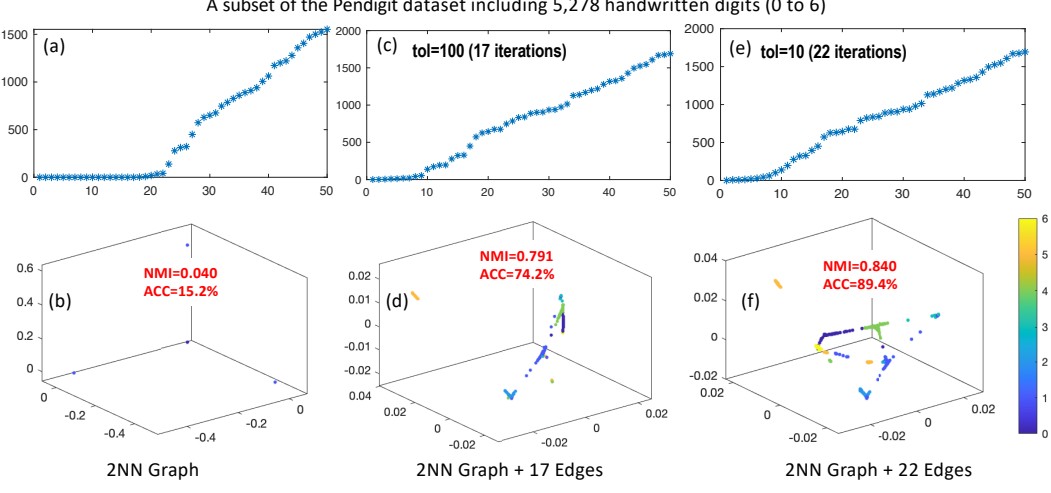

Figure 6: The first 50 Laplacian eigenvalues (top) and 3D spectral drawings (bottom) of the 2NN graph in figures (a) and (b), and the GRASPEL-learned graphs in figures (c) to (f).

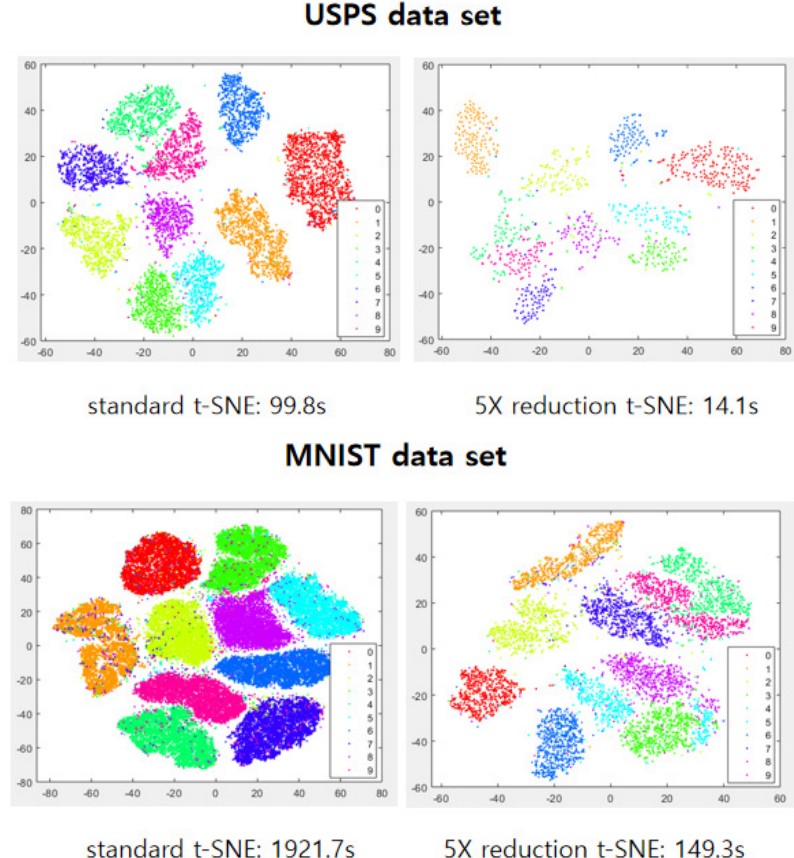

Figure 7: Multilevel t-SNE visualization results.

**Discussion.** The superior performance of GRASPEL is due to the following reasons: **1)** In traditional kNN graphs, all the nodes have the same degrees; as a result, the clustering may strongly favor balanced cut, which may lead to improper cuts in high-density regions of the graph. In contrast, GRASPEL always learns ultra-sparse graphs that only include edges with the largest impact to graph spectral (structural) properties; as a result, the corresponding cuts will always occur in proper regions of the graph, which enables to handle even unbalanced data. **2)** Recent work (Garg et al., 2018) shows the fundamental connections between spectral properties of graphs associated with data and the inherent robustness to adversarial examples. Since GRASPEL identifies candidate edges by leveraging spectral graph properties, the learned graph structure will also be robust to input noises (perturbations).

A.6.2   GRAPH LEARNING FOR DIMENSIONALITY REDUCTION (DR)

The t-Distributed Stochastic Neighbor Embedding (t-SNE) has become one of the most popular visualization tools for high-dimensional data analytic tasks (Maaten & Hinton, 2008; Linderman & Steinerberger, 2017). However, its high computational cost limits its applicability to large scale problems. An substantially improved t-SNE algorithm has been introduced based on tree approximation (Van Der Maaten, 2014). However, for large data set the computational cost can still be very high.

A multilevel t-SNE algorithm has been proposed in (Zhao et al., 2018) leveraging spectral graph coarsening as a pre-processing step applied to the original kNN graph. A much smaller set of representative data points can be then selected from the coarsened graph for t-SNE visualization. In this work, we use GRASPEL to learn sparse graphs that can be further reduced into much smaller ones using spectral graph coarsening. Then more efficient t-SNE visualization can be achieved based on the sampled data points corresponding to the nodes in the coarsened graphs.

In our experiments, we first construct initial graphs by applying a spectral sparsification procedure to the 5NN graphs of both the MNIST and USPS data sets. Then a spectral graph coarsening procedure (Zhao et al., 2018) has been applied to create a hierarchy of coarse-level graphs. The t-SNE visualization can be obtained by directly using the data points corresponding to the nodes on the coarsest graph. Figure 7 shows the visualization and runtime results of the standard t-SNE (with tree-based acceleration) (Van Der Maaten, 2014) and the multilevel t-SNE algorithm (Zhao et al., 2018) based on graphs learned by GRASPEL. When using a $5X$ graph reduction ratio, t-SNE can be dramatically accelerated (12.8X and 7X speedups for MNIST and USPS data sets, respectively) without loss of visualization quality.

