# OpenReview forum: "Graph Learning via Spectral Densification"
_ICLR.cc/2021/Conference — Reject_

### Official Review · AnonReviewer3 · 2020-10-27
**Fast graph learning based on interesting mathematical formulation**

**Rating:** 6
**Confidence:** 3

**Review:**

Summary and significance: Learning a graph from data is an important, yet less studied, problem. The proposed algorithm (GRASPEL) is based on a graphical Lasso formulation with the precision matrix restricted to be a graph Laplacian. The algorithm starts with a sparse kNN graph, and recursively adds critical edges (identification of these critical edges based on Lasso and spectral perturbation analysis is the main contribution of the paper).
The outcome is a highly scalable that learns a graph in nearly linear time (ignoring log factors and number of recursions). The scalability of the algorithm makes the contributions significant.

Originality: The basic formulation and idea of selecting spectrally critical edges seem original and interesting, although the reviewer is not an expert in related methods. The authors should note that the there are graph learning methods based on solution of graphical lasso [Pavez, Ortega, ICASSP 2016; Kumar et al, Neurips 2019]. Beyond this step, the authors employ several existing techniques to make GRASPEL scalable (although this part is not highly novel, the overall method is original).

Quality and clarity: The theory in the paper is technically sound. The only exception (this is also an issue about clarity) is the assumption that $U_N^T e_{pq} \approx U_r^T e_{pq}$. In general, this is not valid and hence it should be clarified when this assumption is reasonable. The experimental section is well executed.
The paper presented is mostly good. It would help to include Algorithm 1 and Table 1 in the main paper.

Typo: On page 6 (Phase A), perhaps the authors meant (r=2) instead of (r=1). Also reference of Carey is currently in all caps

---

> ### Author Response · Authors · 2020-11-17
> **Our response to the reviewer's questions**
>
> "Quality and clarity: The theory in the paper is technically sound. The only exception (this is also an issue about clarity) is the assumption that
> UNTepq≈UrTepq
> . In general, this is not valid and hence it should be clarified when this assumption is reasonable. The experimental section is well executed. The paper presented is mostly good. It would help to include Algorithm 1 and Table 1 in the main paper. "
>
> ----Our response: Thanks very much for the comments. We have substantially improved the draft and included additional results to better demonstrate the benefit of this work. As derived in the latest paper draft, the embedding distortion is defined as the ratio of the spectral embedding distance on the graph and the actual L2 distance in the original data space. Increasing embedding dimension r will allow more accurate estimation of the partial derivative in (12). As r->N and sigma-> inf, the spectral embedding distance becomes the graph resistance distance.

---

### Official Review · AnonReviewer2 · 2020-10-28
**Interesting New Approach to Important Problem**

**Rating:** 6
**Confidence:** 4

**Review:**

This paper studies ways of adding edges to graphs to improve the result of spectral embedding / clustering. It refines existing embeddings using by measuring edges' effect on Laplacian eigenvalues, and adjusting such edges to reduce the distortions. The performance of the algorithm is justified using developments of worst-case efficient algorithms for Laplacian matrices, and experimentally, the algorithm converges quickly when starting with nearest neighbor graphs, and leads to significant increases in accuracy.

Strengths:
+ the paper puts together a lot of different ideas, many of which have solid theoretical foundations.
+ full experimental evaluation on a moderate sized data set that demonstrates both good results and good performance.

Weaknesses
- the presentation could use significant improvement
- the ideas are a bit disconnected, at least when one try to follow the ideas mathematically.

I find the approach taken by this paper quite interesting: node embeddings are now widely used to preprocess graphs into vector data more friendly to learning pipelines. Many of these methods add additional edges imperatively / via local methods, without taking the overall data set into account. Iterating this process based on the overall embedding is an interesting, but algorithmically more intensive approach. To carry this out, the authors combined a variety of interesting tools, as well as some high performance packages that they developed. I feel the novelty of this combination, as well as the gains obtained, should make this result of broad interest to those working on spectral clustering/embeddings. However, many of the other reviews point out gaps in both the ideas and presentation, and I'm inclined to agree with them that this paper can benefit from a thorough revision before appearing at a major conference.

---

> ### Author Response · Authors · 2020-11-17
> **Thanks for the review comments**
>
> "This paper studies ways of adding edges to graphs to improve the result of spectral embedding / clustering. It refines existing embeddings using by measuring edges' effect on Laplacian eigenvalues, and adjusting such edges to reduce the distortions. The performance of the algorithm is justified using developments of worst-case efficient algorithms for Laplacian matrices, and experimentally, the algorithm converges quickly when starting with nearest neighbor graphs, and leads to significant increases in accuracy.
> I find the approach taken by this paper quite interesting: node embeddings are now widely used to preprocess graphs into vector data more friendly to learning pipelines. Many of these methods add additional edges imperatively / via local methods, without taking the overall data set into account. Iterating this process based on the overall embedding is an interesting, but algorithmically more intensive approach. To carry this out, the authors combined a variety of interesting tools, as well as some high performance packages that they developed. I feel the novelty of this combination, as well as the gains obtained, should make this result of broad interest to those working on spectral clustering/embeddings."
>
> ----Our response: Thanks very much for the comments. We have substantially improved the draft and included additional results to better demonstrate the benefit of this work.

---

### Official Review · AnonReviewer4 · 2020-10-28
**Nice idea but performance relies on a number of other techniques + presentation can be improved**

**Rating:** 5
**Confidence:** 4

**Review:**

Update: In the revised version, the authors have addressed some of my technical concerns, therefore I slightly increased my score.


Summary:

The paper proposes a method to learn a sparse graph from data by optimizing an objective similar to the graphical Lasso over the set of valid Laplacian matrices. The method starts with an initial sparse kNN graph which has been further sparsified using spectral sparsification algorithms. It then applies an approach referred to as "spectral graph densification", where spectrally-critical edges are identified which have large embedding distortions and thus significantly impact the objective function. The approach is terminated based on spectral stability checking, where the iterations are terminated when the embedding distortions become sufficiently small.


Strengths:

- The paper proposes a new approach for learning sparse graphs from data.
- Compared to prior spectral graph sparsification algorithms which remove edges from a given graph, the proposed approach starts with a sparse graph and adds edges into the graph iteratively. It is shown in the experiments that this approach leads to a fast improvement in the objective function values in a few iterations.
- The proposed approach has running time in $O(N \log N)$ instead of $O(N^2)$ for each iteration and is thus faster than previous graph learning methods.


Concerns:

- The discussion in the beginning of Section 3 seems to suggest that the formulation of the objective (4) is an individual contribution of the paper. However, the convex optimization problem in (4) has been previously proposed in [Lake & Tenenbaum, 2010] (see also the discussion in [Dong et al, 2019]). The paper mentions [Dong et al, 2019] but also a reference to [Lake & Tenenbaum, 2010] should be provided.
- The proposed approach relies on a number of other techniques: it requires as input a kNN graph which is then further sparsified using other spectral sparsification algorithms ([Spielman & Srivastava, 2011, Feng, 2018]) and then a lower-dimensional vector representation is computed by using a nearly-linear-time spectral graph embedding procedure ([Zhao et al, 2018]). The only technical contribution of the paper is then to provide a greedy optimization scheme for the objective in (4) by selecting the spectrally critical edges.
- It is stated in the introduction that the experimental results show that the graphs learned by the proposed technique can lead to more efficient and accurate spectral clustering (SC) as well as dimensionality reduction. However, the results for spectral clustering and dimensionality reduction are only in the appendix and not discussed in the main part of the paper. In order to make the paper self-contained, at least parts of these results should be moved to the main paper. To make space for these results, one suggestion is to merge the sections 3.3 and 4 into one as they are a bit redundant.


Minor Comments:

- After (4) it says that "It can be shown that the three terms in (4) are corresponding to $\log \det (\Theta)$,  $Tr(\Theta S)$ and $\beta||\Theta||_1$ in (3), respectively". What is meant by "correspond" in this context?
- In equation (11), what is the index $r$? It has not been defined before.
- As (14) is directly obtained from (12) by plugging (13) into the left hand side of (12), why is there $ \approx$ in (14) instead of $=$ ? Also, what is the purpose of the right hand side of (12) (after the $\geq$)?
- The claim that the spectral embedding distances on the learned graph will encode the $l_2$ distances between the original data points should be further clarified.
- Theorem 1 is a bit imprecise, as it makes a statement about the "spectral criticality" of a candidate edge, which has not been formally defined before (spectral critical edges are only introduced informally before as "the ones which can most effectively perturb the graph spectral properties"). The proof in the appendix then computes the relative spectral perturbation of the first $r$ eigenvalues due to the inclusion of the edge $(p,q)$ but does not precisely relate it to the statement in the Theorem.
- Is the notion of "graph spectral stability" novel? If not, please provide a reference.



Conclusion:

The idea of using spectral densification to optimize the graph learning objective is nice and elegant. However, I vote for rejection as the performance of the method is reliant on a number of other techniques for the construction of the initial sparse graph and fast computation of the spectral embedding. Moreover, the presentation of the paper could be improved (see comments above).



References:

[Dong et al, 2019] Xiaowen Dong, Dorina Thanou, Michael Rabbat, and Pascal Frossard. Learning graphs from data: A signal representation perspective. IEEE Signal Processing Magazine, 36(3):44–63, 2019.

[Feng, 2018] Zhuo Feng. Similarity-aware spectral sparsification by edge filtering. In 2018 55th ACM/ESDA/IEEE Design Automation Conference (DAC), pp. 1–6. IEEE, 2018.

[Lake & Tenenbaum, 2010] B. Lake and J. Tenenbaum. Discovering structure by learning sparse graph. Proceedings of the Annual Cognitive Science Conference, 2010.

[Spielman & Srivastava, 2011] Daniel Spielman and Nikhil Srivastava. Graph Sparsification by Effective Resistances. SIAM Journal on Computing, 40(6):1913–1926, 2011.

[Zhao et al, 2018] Zhiqiang Zhao, Yongyu Wang, and Zhuo Feng. Nearly-linear time spectral graph reduction for scalable graph partitioning and data visualization. arXiv e-print, arXiv:1812.08942, 2018.

---

> ### Author Response · Authors · 2020-11-17
> **Our response to reviewer's comments**
>
> "Concerns:
> The discussion in the beginning of Section 3 seems to suggest that the formulation of the objective (4) is an individual contribution of the paper. However, the convex optimization problem in (4) has been previously proposed in [Lake & Tenenbaum, 2010] (see also the discussion in [Dong et al, 2019]). The paper mentions [Dong et al, 2019] but also a reference to [Lake & Tenenbaum, 2010] should be provided.
> The proposed approach relies on a number of other techniques: it requires as input a kNN graph which is then further sparsified using other spectral sparsification algorithms ([Spielman & Srivastava, 2011, Feng, 2018]) and then a lower-dimensional vector representation is computed by using a nearly-linear-time spectral graph embedding procedure ([Zhao et al, 2018]). The only technical contribution of the paper is then to provide a greedy optimization scheme for the objective in (4) by selecting the spectrally critical edges."
>
> ----Our response: In the revised paper, (14) implies that including a candidate edge with (a) a larger embedding distortion and (b) a greater embedding distance will allow a faster convergence of (4) according to the gradient descent (GD) method for solving convex problems. The most spectrally-critical edge can therefore be identified using the following two phases in Step 3: a large embedding distance can be guaranteed by limiting the search within candidate edges connecting between the top and bottom few nodes sorted by the Fiedler vector; subsequently, only the candidate edge with the largest embedding distortion will be added into the latest graph. As for validation of the approach, we provide additional experimental results in Figure 4 to show the significantly improved effective-resistance distance correlation with the original data set.
>
> "It is stated in the introduction that the experimental results show that the graphs learned by the proposed technique can lead to more efficient and accurate spectral clustering (SC) as well as dimensionality reduction. However, the results for spectral clustering and dimensionality reduction are only in the appendix and not discussed in the main part of the paper. In order to make the paper self-contained, at least parts of these results should be moved to the main paper. To make space for these results, one suggestion is to merge the sections 3.3 and 4 into one as they are a bit redundant."
>
> ----Our response: In the revised paper, we have added more results, which does not allow us to include all important ones into the main section. But the most interesting results have been provided in the main section with detailed discussions.
>
> "Minor Comments:
> After (4) it says that "It can be shown that the three terms in (4) are corresponding to ....  but does not precisely relate it to the statement in the Theorem.
> Is the notion of "graph spectral stability" novel? If not, please provide a reference."
>
> ----Our response: In the revised paper, we clearly stated that the idea of checking convergence based on graph spectral stability is novel (page 2). Regarding other concerns, we have provided significantly improved writing and results to avoid all potential confusions.
>
> "Conclusion:
> The idea of using spectral densification to optimize the graph learning objective is nice and elegant. However, I vote for rejection as the performance of the method is reliant on a number of other techniques for the construction of the initial sparse graph and fast computation of the spectral embedding. Moreover, the presentation of the paper could be improved (see comments above)."
>
> ----Our response: This is the first work introducing a truly scalable method for estimating attractive GMRFs based on latest spectral graph algorithms. The spectral embedding (or approximate effective-resistance) distances on the graph will encode the similarities between the original input data points. In the revised draft, we have further highlighted the novelties of this work, and removed the parts that may cause confusions.

---

### Official Review · AnonReviewer1 · 2020-10-28
**Reviewer 1: Graph Learning via Spectral Densification**

**Rating:** 5
**Confidence:** 3

**Review:**

The paper proposes a graph learning method for spectral embedding and associated problems such as clustering and dimension reduction. What differentiates the method from much the existing literature is that it focuses approximating an optimal densification of a very sparse initial graph rather than on sparsification of an initial graph, as is more common. The method is based on iteratively identifying edges to add to the graph so as to best improve the corresponding spectral embedding, so called "spectrally critical" edges. The authors motivate spectral criticality in relation to the partial derivatives of an objective function inspired by the log-likelihood of a Gaussian graphical model. In particular, those with the highest partial derivatives will tend to be those which, through their addition to the graph, lead to the greatest increase in this objective. The authors go on to discuss a close connection between spectral criticality and distance distortion when comparing the spectral embedding and the original input space. Since the initial graph is very sparse it can be efficiently determined, and the relatively small number of additional edges which need to be added by the proposed method to obtain a high quality embedding means that the entire procedure can be implemented efficiently.

The paper is well written for the most part, and the method is intuitive and persuasive. The connection between the partial derivatives and the distance distortions is very pleasing and provides a commonsense interpretation of the operations of the method. In addition the empirical performance of the method on some important benchmarks seems to be good.

My main concern about the paper is how the method is presented. The connection to the graphical LASSO seems at first to be pivotal, but then the LASSO component is dropped (\beta set to 0, or effectively ignored in the actual algorithm). Without this connection, then, it isn't clear what is the motivation for the objective being optimised? It seems far more natural to me to motivate the method from the point of view of the distance distortions being used to select edges, and to treat the connection to the log-likelihood objective as an interesting theoretical point. Another connection beyond the graphical LASSO is given in relation to Bregman distances, but there is far too little discussion given for this connection to motivate the use of the objective. Furthermore, although I genuinely do appreciate the motivation of distance preservation, when we think about manifold learning it is only really the local distance structure which is of great importance, and so some of the justification is lost. It is intuitively the case that since the distance distortions are determined as ratios that this will implicitly pick up on deviations in smaller distances over larger ones in any case, so this is practically not a problem, but the authors do not mention this fact and so the direct justification for why this approach works for manifold learning is somewhat lacking. Finally, while the empirical performance shows some promise of the method, it unfortunately leaves a few important questions unanswered. Notably, if we consider Figure 2 we see that the proposed approach rapidly improves on the 2NN graph, and "converges" after relatively few iterations. Using the authors' interpretation of "convergence" in this case, however, it looks like their method does not improve appreciably on the uNN graph. Furthermore the initial uNN graph is far superior in objective value to the modified 2NN graph. I understand that the uNN graph used for initialisation may be far denser than the final GRASPEL graph which started from the 2NN graph, but this isn't discussed, nor is it clear that applying GRASPEL to the 2NN graph is computationally superior to just starting with the uNN solution and not using GRASPEL at all.

The paper and the proposed method clearly have some strong points, however in its current form I am concerned it leaves too much not adequately clear for the reader.

In addition to the points above, below find a few minor comments/questions/typos.
- How does the discussion of smooth signals on graphs connect to the proposed approach? Algebraic connectivity seems like a more natural connection to the way the method is posed.
- what are "attractive" Gaussian Markov Random Fields?
- typo pg 2 "a undirected" -> "an undirected"
- I'm not used to seeing matrices divided by scalars. While it isn't ambiguous, I'd recommend \frac{1}{\sigma^2}I as opposed to \frac{I}{\sigma^2}
- I am confused by the dimensionality. You mention X is M observations on N data entities, which I interpret as sample size  = N and dimension = M. But this doesn't match the description of the sample covariance matrix formulation in the footnote on pg 3. It is also discussed that one makes M i.i.d. observations for this connection to a covariance matrix.
- Why are the embedding distortions proportional to M and not M/r? The latter makes better intuitive sense since we would expect distances to scale roughly with the square root of dimensionality.
- typo (?) pg 4: "there exists no edge with \eta > 1 can be found..."
- In Phase A of step 3 why only search the extreme points in the first non-trivial eigenvector? It seems critical points can show up in subsequent eigenvectors as well
- Apologies if I missed this point, but is it the case that the columns of the data matrix are scaled to have unit norm? If not, wouldn't it be that the distortion distances could be arbitrarily large, since the eigenvectors are normalised?
- In the experimental setup, is it the case that you sample 1/1000 edges which connect two points which lie in the extrema of the Fiedler vector? this is an extremely small number. Not a problem, but if the Fiedler vector is clearly indicative of criticality, why not simply use fewer than 5% each end and look at all pairwise distances among these potentially critical points.
- The discussion on page 7 relating to Figure 2: "As observed in Figure 2 (a) achieves a much greater objective function value after 30 iterations when comparing with (b)" Doesn't the figure show the opposite of this? The light blue line is (a) and the red line is (b)

#########################################################

Final Recommendation:
I have considered the authors' responses to my comments, as well as the assessments given by other reviewers. I still feel as though, while the method looks as though it may offer a potentially useful practical alternative to other graph learning methods, in its current form I do not think it is presented in a manner which warrants acceptance at a prestigious conference such as ICLR. If the decision overall is that the paper is not to be accepted, then I wish the authors well with their work and hope they take into consideration the comments of the reviewers as I do believe the work has potential.

---

> ### Author Response · Authors · 2020-11-17
> **Our response to the reviewer's issues**
>
> "The connection to the graphical LASSO seems at first to be pivotal, but then the LASSO component is dropped... as an interesting theoretical point. "
>
> ----Our response: In the revised paper, we provide a more clearer explanation: since the last two terms in (8) are all fixed (constant) values for a given data matrix X where β can be considered as an additional offset added to all data pairs (candidate edges), we can drop the third term by simply setting β= 0, which will not impact the ranking of candidate edges in graph learning. The above simplification implies  the second  term alone  will effectively penalize graph density  for  estimating Laplacian-like precision matrix: including more edges will result in greater trace Tr(X^T \Theta X).
>
> "Another connection beyond the graphical LASSO ... is somewhat lacking. "
>
> ----Our response: In the revised paper, (14) implies that including a candidate edge with (a) a larger embedding distortion and (b) a greater embedding distance will allow a faster convergence of (4) according to the gradient descent method for solving convex problems. The most spectrally-critical edge can therefore be identified using the following two phases in Step 3: a large embedding distance can be guaranteed by limiting the search within candidate edges connecting between the top and bottom few nodes sorted by the Fiedler vector; subsequently, only the candidate edge with the largest embedding distortion will be added into the latest graph. As for validation of the approach, we provide additional experimental results in Figure 4 to show the significantly improved effective-resistance distance correlation with the original data set.
>
> "Finally, while the empirical performance shows ... just starting with the uNN solution and not using GRASPEL at all."
>
> ----Our response: In the revised paper, we provide a better discussion about using 2NN or uNN as the initial graph for GRASPEL iterations in Section 5.2: “starting with an uNN achieves a greater objective function value when comparing with 2NN, which is mainly due to a much sparser graph structure enabled by spectral graph sparsification. However, it may be  challenging to choose a reasonably good k for creating the initial kNN graph that will be further spectrally sparsified for the following GRASPEL iterations: choosing a too large k will result in very dense graph even after spectral sparsification, whereas choosing a small k for sparsification may lead to slower convergence.”
>
> "I am confused by the dimensionality. ..."
>
> ----Our response: Thanks for the suggestion. In the revised paper, we have provided a much better explanation regarding the above issue. “For example, the USPS data set including 9,298 images of handwritten digits with each image having 256 pixels will result in an N X M feature matrix X where N=9,298 and M=256.”
>
>
> "Why are the embedding distortions proportional to M and not M/r? ..."
>
> ----Our response: as derived in the latest paper draft, the embedding distortion is defined as the ratio of the spectral embedding distance on the graph and the actual L2 distance in the original data space. Increasing embedding dimension r will allow more accurate estimation of the partial derivative in (12). As r->N and sigma-> inf, the spectral embedding distance becomes the graph resistance distance.
>
> "typo (?) pg 4: "there exists no edge with \eta > 1 can be found... eigenvectors as well"
>
> ----Our response: Thanks for the suggestions. Searching in 1D embedding space is more efficient, while using higher dimensions will involve more complicated searching methods. More importantly, we use higher spectral embedding space when computing the spectral distortion of each edge, which has been described in Phase B of Step 3 on page 6.
>
> "Apologies if I missed this point, but is it the case that the columns ... are normalised?"
>
> ----Our response: Thanks for the suggestions. In the footnote of page 3, we described the normalization of feature matrix X.
>
> "In the experimental setup,... critical points."
>
> ----Our response: Thanks for the suggestions. We have fixed the confusions in the latest revised draft. In Section 5.1, we added “Note that choosing a smaller epsilon value will allow more effective edge sampling for estimating more global  graph (manifold) structural properties, while choosing a greater epsilon  value will require more   samples but lead to better   preservation of mid-to-short range graph (manifold) structural properties.”
>
> "The discussion on page 7 relating to Figure 2: ... the red line is (b)"
>
> ----Our response: Thanks for the suggestions. We have fixed the confusions in the  latest revised draft. Note that choosing a smaller epsilon value will allow more effective edge sampling for estimating more global  graph (manifold) structural properties, while choosing a greater epsilon  value will require more   samples but lead to better   preservation of mid-to-short range graph (manifold) structural properties.

---

> > ### Comment · AnonReviewer1 · 2020-11-23
> > **Still not totally convinced**
> >
> > while the response from the authors (along with the updated paper) have addressed some of my concerns, I still have a problem with the presentation of the method from the point of view of the graphical LASSO. While I agree that the selection of "next edge to add" is independent of the setting of \beta, that was not the problem that I had. The problem is that by setting \beta to zero there is no LASSO component, it is simply the log-likelihood for the Gaussian PGM. How the method seems rather to work is simply an approximate co-ordinate descent. This is in fact far closer connected to the forward stagewise learning, which has theoretical connections with the LASSO itself, but is far from equivalent.
> >
> > I am still very borderline on the paper, and would welcome input from other reviewers, some of whom are very positive about it.

---

> > > ### Author Response · Authors · 2020-11-23
> > > **A revised paragraph regarding the general stagewise algorithm**
> > >
> > > Thanks very much for your suggestions and comments. The proposed GRASPEL algorithm shares some common features of the general stagewise algorithm discussed in a recent paper "A General Framework for Fast Stagewise Algorithms". We have added the following to the paragraph  under "the proposed approach" on page 5:
> > > "We note that the proposed GRASPEL algorithm shares some similar features as the general stagewise algorithm: as the step size goes to zero the sequence of forward stagewise estimates will exactly coincide with the lasso path. Consequently, the GRASPEL algorithm will produce an approximate solution to the original graphical Lasso problem when using a rather small step size (e.g. adding only one edge with a small edge weight in each GRASPEL iteration)."

---

### Author Response · Authors · 2020-11-17
**The revised paper has addressed major concerns from the reviewers**

In the latest draft, we have substantially improved the writing to make each technical component more clearly explained/discussed according to reviewers' comments and suggestions. We highlighted the original contribution of this work. Also, we discussed how to set up the input parameters for running the GRASPEL algorithm.

Regarding experimental results, we have added Figure 4 to show the correlation between effective-resistance distances and the L2 distances among the original data points using the graphs learned/constructed by different methods. For the spectral clustering application, additional graph density results have also been provided. We also decompose the original big table into multiple smaller tables to more clearly demonstrate the results.

---

### Decision · Program_Chairs · 2021-01-07
**Final Decision**

**Decision:**

Reject

**Comment:**

The reviewers generally like the paper, in particular the scalability of the proposed approach. The author response and revised version clarified some questions of the reviewers, however, it didn't fully mitigate their concerns.